# Investigating the Effects of Polypropylene Fibers on the Mechanical Strength, Permeability, and Erosion Resistance of Freshwater and Seawater Mixed Concretes

Thamer Alomayri [1,*], Babar Ali [2,*], Syed Safdar Raza [3], Hawreen Ahmed [4,5,6] and Moustafa Hamad [7,8,*]

1    Department of Physics, Faculty of Applied Science, Umm Al-Qura University, Makkah 21955, Saudi Arabia
2    Department of Civil Engineering, COMSATS University, Islamabad 57000, Pakistan
3    Department of Civil Engineering, Faculty of Engineering, Bahauddin Zakariya University,
     Multan 66000, Pakistan; safdarshah91@bzu.edu.pk
4    Department of Highway and Bridge Engineering, Technical Engineering College, Erbil Polytechnic University,
     Erbil 44001, Iraq; hawreen.a@gmail.com
5    Department of Civil Engineering, College of Engineering, Nawroz University, Duhok 42001, Iraq
6    Civil Engineering, Architecture and Georesources Department, Instituto Superior Técnico, Universidade de
     Lisboa, Av. Rovisco Pais, 1049-001 Lisbon, Portugal
7    Department of Civil Engineering, College of Engineering, Al Baha University,
     Al Baha P.O. Box 1988, Saudi Arabia
8    Civil Engineering Department, Faculty of Engineering, Delta University for Science and Technology,
     Belkas P.O. Box 11152, Egypt
*    Correspondence: tsomayri@uqu.edu.sa (T.A.); babar.ali@scetwah.edu.pk (B.A.); m.saad@bu.edu.sa (M.H.)

**Abstract:** Seawater mixed (SW) concrete lessens the freshwater (FW) demand and eases the stress on the already depleting FW resources. The use of SW concrete is a sustainable solution that mitigates the environmental impact of concrete production, especially in coastal regions and islands vulnerable to FW scarcity. This study investigated the influence of polypropylene (PP) fiber incorporation on high-performance-SW concrete's long-term mechanical and durability performance. The findings indicate that the incorporation of seawater in the production of concrete containing ground granulated blast furnace slag (GGBFS) has a beneficial effect on its early strength. This is due to the fact that SW accelerates the hardening process. SW concrete mixes showed an improvement in strength with aging. The difference between the strength of SW and FW concretes reduced with aging. The PP fiber showed phenomenal improvements in the tensile properties of SW and FW concretes. At the addition of 0.3% PP fiber, SW yielded 56% and 48% higher splitting tensile and flexural strength than plain FW concrete at 28 days, respectively. The use of 0.15% of PP fiber caused notable reductions of around 20% in the water absorption (WA) capacity and a 12–20% reduction in chloride ion permeability (CIP) of SW concrete. The incorporation of PP fiber increases the number of drying–wetting cycles to initiate the erosion of SW and FW concretes in a simulated environment. The use of 0.15% PP fiber is beneficial, as compared to 0.3% PP fiber to control the tidal erosion of SW and FW concretes. After exposure to 126 drying–wetting cycles (stimulated tidal erosion), the mass loss of SW concrete was reduced from 0.56% to 0.22%.

**Keywords:** tidal environment; simulated environment; tensile reinforcement; sulphate resistant cement; fiber technology

## 1. Introduction

Minimizing the environmental impact of concrete involves adopting practices, such as using alternative materials and implementing efficient production methods to reduce carbon emissions and resource consumption [1]. Seawater mixed (SW) concrete is becoming more popular due to its many benefits, particularly in coastal areas where FW is limited and expensive. The use of this type of concrete can help in lowering the demand for freshwater

(FW), resulting in a reduced strain on FW resources. SW concrete is a sustainable option for construction projects, contributing to the development of a sustainable and efficient future. To ensure its efficacy, it is important to assess the properties of SW concrete in terms of its long-term durability, weathering resistance, and mechanical performance [2]. Based on the existing literature [3,4], it appears that the mechanical properties of SW concrete are not notably different from those of the FW-mixed concrete. Narver [5] showed that during the first month, the compressive strength of SW concrete was higher than that of FW concrete. Nevertheless, after 3 months, the strength of SW concrete decreased by 6% compared to FW concrete. Steinour's research [6] similarly found that SW concrete showed an improvement in early strength. SW usage as a mixing water has an accelerating effect similar to the addition of salt-based chemical accelerator, such as calcium chloride [7]. However, over time, the strength of SW concrete decreased by 8–15% compared to FW concrete. Recent findings also confirm the similar results regarding the positive effect of SW mixing on the early strength and a minor degrading impact on the later strength of concrete [8,9]. The salt content in SW concrete usually has a declining effect on the properties of fresh concrete, resulting in reduced workability and quicker setting [10–12]. However, certain studies [12,13] suggest that the appropriate use of chemicals and mineral admixtures (metakaolin, fly ash, GGBFS) can improve the workability of SW concrete.

In the case of traditional FW concrete application in steel-reinforced structures for marine construction, the chlorides in marine environment can remove the passive layer of steel and initiate the corrosion [14]. It is essential to control the chlorides penetration into the concrete to save the reinforcing steel from the chloride-related corrosion. One main line of defence is to use concrete with high penetration resistance against the chlorides. Different mineral admixtures (i.e., silica fume, GGBFS, fly ash, and metakaolin) can effectively advance the resistance of concrete's chloride ion permeability [15]. SW mixing is not suitable for concrete with steel reinforcement. SW mixing is suitable for plain concrete applications and structural concrete reinforced with fiber-reinforced polymer rebars [16,17].

The other concern for SW and FW concretes in marine environment is their long-term durability and resistance to erosion. Concrete structures exposed to the marine environment are subjected to extremely severe and corrosive conditions. SW has a high concentration of erosive elements, such as magnesium salt, chloride salt, and sulphate, which can quickly deteriorate the concrete structure [18]. Several factors can contribute to the complex process of chemical erosion of concrete. Exposing concrete structures to tidal erosion and wet–dry cycles can exacerbate their durability concerns. As a result, concrete structures that are exposed to tides generally have a limited lifespan of fewer than 10 years [19,20]. According to Li et al. [21], the erosion of cement-based materials caused by individual ions in SW is distinct from erosion that takes place over a longer period. Moreover, the erosion process is intensified by the combination of magnesium ions ($Mg^{2+}$) and sulphate ions ($SO_4^{2-}$). Cheng et al. [22] conducted a test where they exposed mortar to cycles of drying and wetting to investigate how various ions impact the material's physical degradation. The test results showed that the coexistence of sulphate and magnesium ions in a chloride environment caused an increase in the porosity of the mortar, resulting in an accelerated erosion process. An experimental study [23] found that the presence of chloride in SW accelerates the cement hydration process, leading to a higher level of autogenous shrinkage in concrete mixed with SW, when compared to conventional concrete. Li et al. [24] found that after 1 year of conditioning in SW, concrete mixed with SW reduced around 7% of its compressive strength.

Previous studies [25,26] have suggested that the addition of mineral admixtures to SW concrete can improve its performance to some degree. Additionally, this technique can provide several technological and economical benefits by enhancing the micro-level characteristics of the concrete and reducing the amount of cement needed. According to Yi et al. [27], adding fly ash, GGBFS, and silica fume (SF) as a partial replacement of Portland cement can enhance the concrete's ability to resist corrosion when subjected to SW. In a study by Jau et al. [28] on the performance of concrete with GGBFS under wet–dry

cycles, it was found that the loss of strength in SW concrete containing GGBFS decreased after being immersed in SW for a period of 90 days. According to Nazanin et al. [29], adding mineral admixtures to SW-based slurry can effectively impede the erosion caused by harmful ions and result in a more compact structure. The study also found that the use of quaternary mineral admixture blending was more effective than ternary blending. Additionally, compared to concrete, the SW-slurry containing mineral admixture had lower porosity, indicating a denser structure.

It is widely acknowledged that the inclusion of fibers in traditional FW concrete can effectively control the spread of cracks and enhance its ductility [30–34]. Although there are different kinds of fibers, such as metallic, glass, synthetic, and natural fibers [35,36], synthetic fibers, such as PP fiber are suitable for SW concrete applications due to their exceptional ability to resist SW and chemical corrosion. Li et al. [24] reported that the high performance and ultra-high performance concretes reinforced with steel fibers retained acceptable strengths and shapes after 1 year of conditioning in SW. However, no particular study is found that explains and compares the tidal erosion behavior of plain and fiber-reinforced SW and FW concretes. No research offers a comprehensive explanation of the mechanical, permeability, and drying–wetting erosion behavior of SW-mixed concrete that has been reinforced with PP fiber. Previous studies [37–39] have shown that PP fiber is capable of controlling the absorption and shrinkage behavior of traditional FW concrete. Therefore, adding PP fiber to SW-mixed concrete can have a beneficial effect on its mechanical and durability properties.

The above premise implies that SW concrete prepared with the Portland cement is vulnerable to erosion and deterioration as it ages. However, the potential solution to this issue is to add fibers and GGBFS to the SW concrete mixture. This could lead to an improvement in the properties of the SW concrete and potentially address the problems associated with the application of SW concrete. The study aimed to examine how the inclusion of PP fiber would impact the mechanical and durability properties of HPC. The HPC was created using SW and a binary cement-GGBFS binder. In the experiment, GGBFS was utilized to substitute 50% of Portland cement. The study involved adding PP fiber at a volume fraction of 0.15% and 0.3% in both FW and SW concretes. In order to assess the mechanical characteristics of the mixes, tests were conducted to determine the splitting-tensile strength, flexural strength, and compressive strength at different ages. The durability of the mixes was assessed by analyzing their water absorption (WA) capacity, chloride ion permeability (CIP), and erosion characteristics during the drying–wetting cycles.

## 2. Materials and Methods

### 2.1. Constituent Materials

The type of cement used was Portland cement which met the standards specified in ASTM C150 for general usage [40]. In all concrete mixes, both GGBFS and Portland cement were used as the binders. The quenched molten slag obtained from blast furnace was sourced from a steel mill located in Lahore, Pakistan, and had a texture similar to that of stone. It was processed through fine grinding until it reached a size of 75 μm, which was considered appropriate for use as supplementary cementitious material. Both the physical and chemical properties of the GGBFS and Portland cement samples are detailed in Table 1.

For the fine aggregate, the study made use of siliceous sand obtained from the Lawrancepur quarry located in Attock, Pakistan. The sand sourced from the Lawrancepur quarry had a fineness modulus of 2.9, which is considered appropriate for producing high performing concretes in Pakistan. Dolomitic coarse aggregate was sourced from the Kiraana-Hill quarry situated in Sargodha, Pakistan. The research employed coarse aggregates, which had a maximum size of 12.5 mm, and fine aggregates, which had a maximum size of 4.75 mm. Table 2 outlines the crucial characteristics of the aggregates that were utilized in the study. Figure 1 displays the gradation of aggregate particles used in the study. The gradation charts adhere to the ASTM C33 [41] standards' upper and lower limits.

**Table 1.** Properties of cement and GGBFS.

| Characteristics | | GGBFS | Portland Cement |
|:---:|:---:|:---:|:---:|
| **Chemical** | CaO (%) | 35.5 | 62.6 |
| | SiO$_2$ (%) | 33.7 | 19.2 |
| | Al$_2$O$_3$ (%) | 13.4 | 5.6 |
| | Fe$_2$O$_3$ (%) | 0.8 | 2.7 |
| | MgO (%) | 6.0 | 1.3 |
| | SO$_3$ (%) | 1.1 | 3.4 |
| | LOI (%) | 0.5 | 1.5 |
| **Engineering** | Initial setting time (h) | - | 3.2 |
| | Final setting time (h) | - | 5.3 |
| | Particle density (g/cc) | 2.95 | 3.11 |
| | Fineness (m$^2$/kg) | 456 | 347 |

**Table 2.** The properties of aggregates relevant to concrete technology.

| Aggregate | Material | Dry Compact Density (kg/m$^3$) | Specific Gravity | 24 h WA (%) |
|:---:|:---:|:---:|:---:|:---:|
| Fine | Siliceous | 1595 | 2.66 | 1.23 |
| Coarse | Dolomite-sandstone | 1684 | 2.69 | 1.34 |

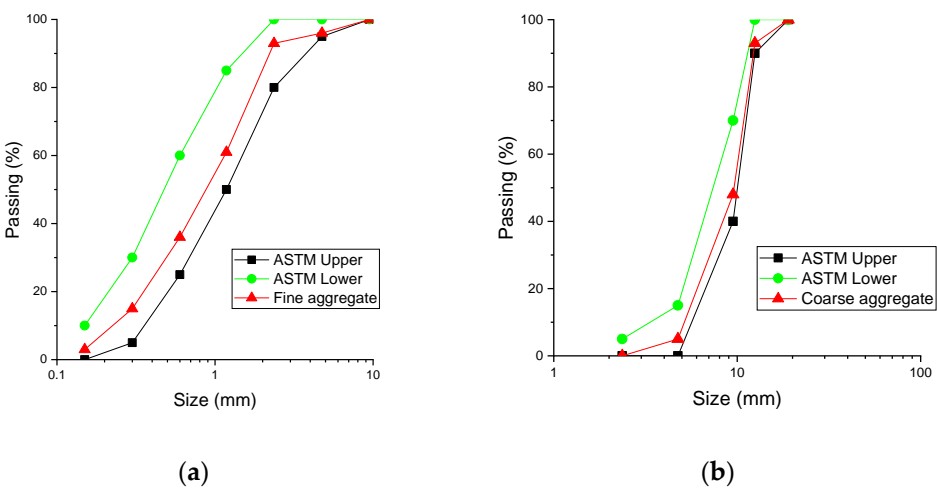

(**a**)  (**b**)

**Figure 1.** Particle size distribution or gradation of (**a**) fine aggregate and (**b**) coarse aggregate.

The study used the PP fiber reinforcement with a length of 12 mm and a diameter of 0.03 mm. The PP fiber material exhibited a tensile strength of 500 MPa and a modulus of elasticity of 5 GPa. The PP fiber sample is shown in Figure 2. The concrete laboratory utilized tap water as a source of FW for the mix preparation in FW series, while the SW was collected from the coast of Hoax Bay, Karachi, Pakistan. Table 3 provides a contrast/comparison between the chemical compositions of FW and SW, which indicates that SW contains relatively higher concentrations of alkali metals, chlorides, sulphates, and sodium than FW. As HPC concrete was used in this study, Viscocrete 3110, a type of polycarboxylate-based third generation superplasticizer, was utilized to achieve the required workability.

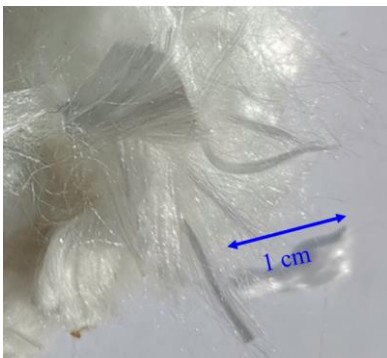

**Figure 2.** The overview of PP fiber reinforcement.

**Table 3.** Composition of FW and SW (%).

| Ions (%) | Ca | Mg | Na | K | B | Cl | $SO_4^{-2}$ |
|---|---|---|---|---|---|---|---|
| FW | 0.012 | 0.002 | 0.005 | 0.001 | 0 | 0.005 | 0.011 |
| SW | 0.047 | 0.141 | 1.171 | 0.043 | 0.001 | 2.071 | 0.284 |

*2.2. Sample Preparation*

Using information from a prior study [42], a standard or control HPC was produced with a w/b ratio of 0.32. Table 4 provides a list of the precise components utilized in the mixture preparations with FW and SW. In summary, the study resulted in the creation of two families with FW and SW. Each family consisted of one plain concrete (P0/FW and P0/SW) and two fiber-reinforced concretes (P0.15/FW, P0.15/SW, P0.5/FW, and P0.5/SW). Every mix consisted of a comparable quantity of fine and coarse aggregates. In both FW and SW series, 50% of Portland cement was substituted with GGBFS. Based on previous research [43,44], PP fiber was added to the mixture at a volume fraction of 0.15% and 0.3% since this proportion was found to have the most beneficial impact on the compressive strength and splitting-tensile strength of HPC. In order to achieve the desired workability for all HPC mixes, a slump range of 150 to 250 mm was chosen. The addition of a superplasticizer called "Viscocrete 3110" was implemented in the mixtures to attain the targeted slump value. To reach the aimed slump value, a superplasticizer dosage of 0.3% wt. of binder was used for FW mixes, while SW mixes were modified with 0.45% dosage. The slump reduced with the replacement of FW with SW due to the higher solid content of the latter as compared to that of the former.

**Table 4.** Concrete mixes.

| Mix IDs | Mixing Water | PP Fiber (%) | OPC (kg/m$^3$) | GGBFS (kg/m$^3$) | Sand (kg/m$^3$) | Coarse Aggregate (kg/m$^3$) | Water (kg/m$^3$) | SP (kg/m$^3$) | PP Fiber (kg/m$^3$) |
|---|---|---|---|---|---|---|---|---|---|
| P0/FM | | 0 | 275 | 260 | 645 | 1090 | 185 | 1.60 | 0.00 |
| P0.15/FM | FW | 0.15 | 275 | 260 | 643 | 1088 | 185 | 1.60 | 1.37 |
| P0.3/FM | | 0.3 | 275 | 260 | 641 | 1086 | 185 | 1.60 | 2.73 |
| P0/SM | | 0 | 275 | 260 | 645 | 1090 | 185 | 2.40 | 0.00 |
| P0.15/SM | SW | 0.15 | 275 | 260 | 643 | 1088 | 185 | 2.40 | 1.37 |
| P0.3/SM | | 0.3 | 275 | 260 | 641 | 1086 | 185 | 2.40 | 2.73 |

The process was divided into three consecutive phases. To begin the concrete mixing process, the binding materials and aggregates were mixed together in a laboratory-grade tilting mixer for 4 min. During the first phase, no water was added. During the second phase of the mixing process, the necessary amount of water and superplasticizer was added to the dry mixture gradually in two separate stages. The mixing process continued for an

additional 4 min after the addition of water and SP to the dry mixture. The final stage of the mixing process involved the gradual addition of the required amount of fiber to the fresh concrete mixture. The final mixing stage was continued for an additional 4 min. After the completion of the mixing, the freshly mixed concrete batches were tested for slump use in accordance with ASTM C143 [45]. In order to shape the fresh concrete into samples, the mixes were placed into steel molds of different shapes, including cubes, cylinders, and prisms, and then compacted using a high-frequency vibrating table. All concrete mixes underwent a uniform vibration process lasting for 30 s. Once the specimens were cast, they remained indoors at room temperature for 24 h to allow them to cure. After the first 24 h of initial setting, the specimens were submerged in tap water for the curing process, and the water temperature was retained at $25 \pm 3\ ^\circ C$.

### 2.3. Determination of Properties

### 2.3.1. Determination of Mechanical Properties

The ASTM C39 method was employed to evaluate the compressive strength of the mixes, which involves testing the compression of cubic samples with 100 mm sides [46]. Compressive strength was measured at four different time points: 7, 28, 91, and 182 days. The ASTM C1609 [47] standard was utilized to measure the flexural strength of plain and fiber-reinforced concrete samples at 28 and 91 days. To determine the flexural strength, the specimens with a width of 100 mm, height of 100 mm, and length of 350 mm were subjected to a third-point bending test. The splitting-tensile strength of the concrete mixes was evaluated at 28 and 91 days using cylindrical samples with a diameter of 100 mm and a height of 200 mm, following the ASTM C496 [48].

### 2.3.2. Determination of Durability Properties

The ability of various types of concrete mixes to resist corrosion of steel was evaluated by determining their quick permeability to chloride ions, which is commonly known as rapid chloride ion permeability (CIP) test. In accordance with the ASTM C1202 standard, samples with a height of 50 mm and a diameter of 100 mm were used to carry out the CIP test [49]. To conduct the test, an electrical voltage of 60 DC V was applied to the sample to produce an electric current. The test measured the speed at which chloride ions penetrated through the sample in coulombs. Samples with a diameter of 100 mm and thickness of 50 mm were used to conduct a water absorption (WA) test, in accordance with the ASTM C948 standard [50]. To carry out the WA test, the samples were submerged in water for a period of 24 h, and then weighed to determine the quantity of water they had absorbed.

This study used an accelerated corrosion-dry–wet cycle test method to evaluate how well FW- and SW-mixed concretes can withstand a simulated marine environment. By using the accelerated corrosion-dry–wet cycle test method, the study was able to imitate the effects of extended exposure to harsh marine environments in a shorter time frame, which yielded important findings regarding the durability of the studied concrete mixes [51]. To replicate the drying–wetting tidal conditions typically found in coastal areas, the study used a simulated erosion system that involved immersing the concrete samples in simulated SW for 16 h and then drying them in an 80 $^\circ C$ oven for 8 h. The methodology employed in this study was borrowed from a prior investigation [51]. In total, the study conducted 126 cycles of drying and wetting, with the concrete specimens being weighed after every 14 cycles (namely, 14, 28, 42, 56, 70, 84, 98, 112, and 126) to track their weight changes over time. By adopting this technique, the study was able to observe the changes in weight (%) over time and obtain valuable information regarding the impact of these cycles on the durability of the FW and SW concretes.

## 3. Results and Discussion

### 3.1. Workability of Fresh Concrete

According to the results presented in Figure 3, the use of SW rather than FW in making concrete resulted in a reduction in the fresh concrete's slump. A drop of more

than 100% in the slump was observed when SW replaced FW. This can be attributed to the fact that SW-based concrete has higher density and a higher concentration of dissolved salts, both of which can contribute to a decrease in the concrete's workability. The presence of increased levels of dissolved salts in SW is the primary factor that contributes to the decreased workability of concrete produced using it. These salts in SW have the ability to interfere with the chemical reactions taking place between cement and water, as well as to absorb water from the concrete mixture. This can lead to a decline in the water-cement ratio, which ultimately results in reduced workability of the concrete [52]. In accordance with the research conducted by Younis et al. [9,53], the FW replacement with SW in the concrete led to a reduction of 20% in the initial slump flow. At the same dosage of SP, the workability of SW concrete is lower than that of the FW concrete; therefore, in the case of SW concrete, the dosage of SP was increased from 0.3% to 0.45% in order to achieve the slump value comparable to that of the SW concrete. As can be noted in Figure 3, the difference between the slump value of SW concrete with 0.45% SP and FW concrete with 0.3% SP is insignificant. Incorporating PP fiber into both SW and FW can cause a decrease in the workability. The presence of micro-fibers in the mixture can result in a "balling effect", leading to the formation of clumps of fibers in the binder matrix. These clumps can negatively impact the workability of HPC by reducing it even further. Fiber addition also increases the cohesion and viscosity of fresh concrete, which may also reflect badly on the slump value. For a given concrete type, despite negative effects on workability, PP fiber incorporation up to 0.3% did not require an increased SP dosage to attain the slump with 150–250 mm.

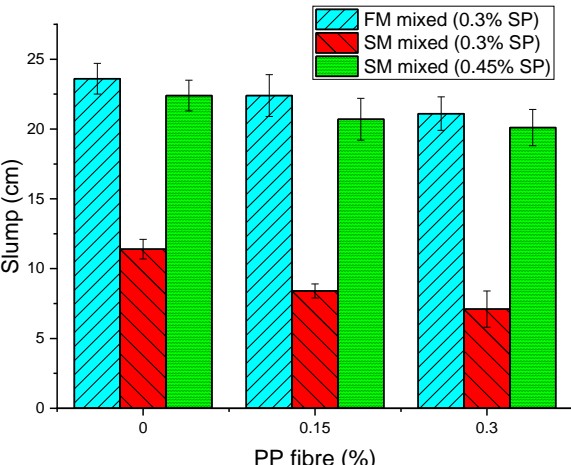

**Figure 3.** Effect of PP fiber incorporation on the slump value of FW and SW concrete mixes.

## 3.2. Compressive Strength

The effect of PP fiber incorporation on the compressive strength of FW and SW concretes is illustrated in Figure 4. Since hybrid binder consisting of GGBFS and OPC was used for the preparation of concrete mix, the early age compressive strength of FW is expected to be low due to the slower rate of pozzolanic reactions, which contribute to the formation of strength gaining C-S-H and C-A-S-H phases over a long period of time [54,55]. Due to the incorporation of 0.15–0.3% PP fiber, the net increments of 3–8% in compressive strength were observed irrespective of the age of testing. PP fibers are commonly used in concrete to improve its mechanical properties, specifically its tensile strength and ductility. However, it is worth noting that PP fiber also has a slightly positive impact on the compressive strength of the concrete. This is due to the fact that the PP fiber dispersed within the concrete matrix acts as a reinforcement, evenly distributing compressive stresses throughout the material and preventing the formation and propagation of cracks. According to Das et al. [56], the inclusion of PP fiber in normal-strength concrete resulted in a slight enhancement of compressive strength of up to 5%, when added at a volume fraction of 0.5%. In this

study, minor improvement in the compressive strength was noted when the fiber content increased from 0.15% to 0.3%.

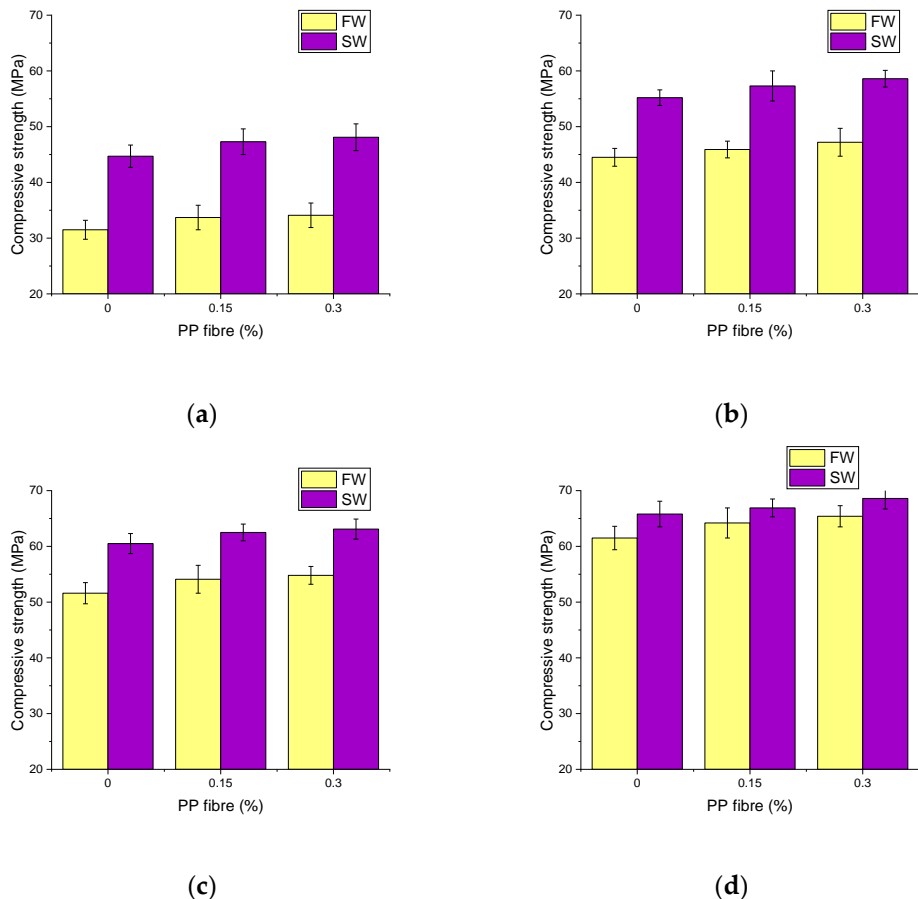

**Figure 4.** Effect of PP fiber on compressive strength of FW and SW concretes measured at (**a**) 7 days; (**b**) 28 days; (**c**) 91 days; and (**d**) 182 days.

The replacement of FW with SW resulted in a significant improvement in compressive strength of concrete, especially at early ages. Up to 40% net increase in 7 days of compressive strength of concrete was observed due to the replacement of FW with SW. While at 28 days, compressive strength of SW concrete was 24% higher than that of the FW concrete. The concrete mixes contain GGBFS, and thus SW showed significant enhancement in their early age compressive strength due to the accelerating effect of Cl- present in SW. Moreover, the increased alkalinity of concrete caused by the presence of alkalis in SW can enhance the reactivity of GGBFS, which may have contributed to the improved compressive strength of the concrete [57]. Due to the presence of sulphate salts in SW, the GGBFS material with high levels of C3A (tricalcium aluminate) can react to form AFm (ettringite) and gypsum. Additionally, when AFm is combined with chloride ions (Cl-), it can form Friedel's salt, which has been shown to improve the density and strength of concrete [58,59]. With the aging, the difference between the compressive strength of FW and SW concretes was reduced. At 91 and 182 days, SW concrete showed 17% and 7% more compressive strength than FW concrete. The difference between SW and FW concretes is reduced with age since GGBFS-OPC binder with FW could contribute to strength at a slower rate, and until 182 days it could contribute to strength up to most of its potential, whereas in the case of SW concrete, the most strength could be developed earlier; therefore, between 91 and 182 days, SW concrete yielded lesser compressive strength than that of the FW concrete.

In contrast to FW concrete, SW concrete could develop micro-cracks due to the crystallization and expansion of salts from SW. This could also lead to a negative impact on the late age strength properties of SW concrete. However, SW concrete generally attained

higher compressive strength values than the FW concretes. Adding PP fiber to concrete had comparable effects on the compressive strength of both SW and FW mixes. The addition of fibers could be particularly useful during the initial stages of salt crystallization, as well as during later stages when micro-cracks may start to form [60]. SW concrete containing 0.3% PP fiber yielded 22% and 12% more compressive strength than FW concrete at 91 and 182 days, respectively.

### 3.3. Splitting-Tensile Strength

The effect of PP fiber content on the splitting-tensile strength of FW and SW concretes is illustrated in Figure 5. It was expected that GGBFS-OPC binder-containing mixes made with FW would yield low splitting-tensile strength at 28 days. It is a widely accepted fact that FW concrete mixes that contain GGBFS take longer to gain strength, as the strength of the concrete is derived from the pozzolanic reactions that occur over an extended period [54]. The addition of PP fiber could significantly improve the early age and late age splitting-tensile strength of FW concrete. At 28 days, FW concrete experienced net improvements of 21% and 33% at 0.15% and 0.3% volume of PP fiber, respectively. Whereas at 91 days, FW concrete showed improvements of 27.3% and 42% at 0.15% and 0.3% volume of PP fiber, respectively. The aging may also improve the net gain in splitting-tensile strength due to fiber addition. Since FW concrete containing hybrid OPC-GGBFS binder developed strength at slower rates, the maturing of the binder matrix at 91 days could significantly increase the efficiency of fiber inclusion. Previously, pore refinement due to the addition of mineral admixtures has been reported to positively influence the tensile-strength efficiency of micro-fiber [61]. Incorporating PP fiber filaments into concrete is beneficial in controlling cracks and enhancing the flexibility of the concrete. This is achieved by the PP fibers absorbing the energy applied to the concrete and dispersing it throughout the concrete matrix, thereby decreasing the likelihood of localized cracking and improving overall splitting-tensile strength. In the event of a crack, the PP fibers act as bridges that distribute the load and prevent further cracking, thereby increasing the structural integrity of the concrete [56,62].

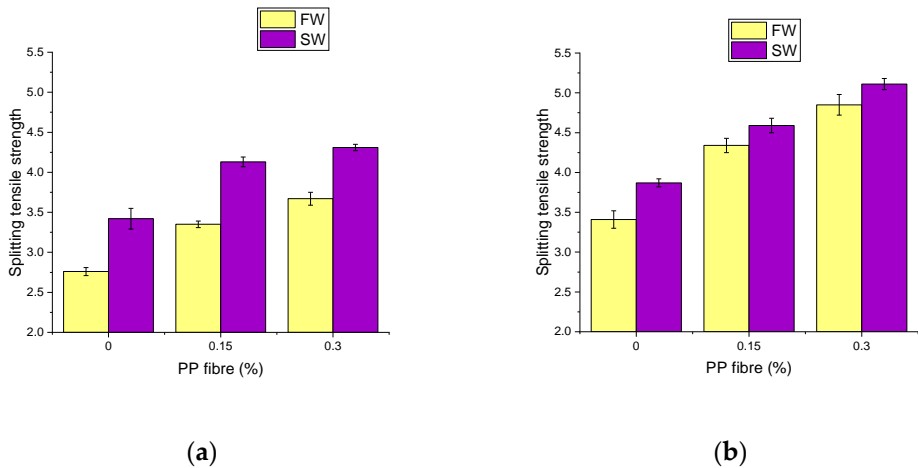

(**a**)                                                                  (**b**)

**Figure 5.** Effect of PP fiber on splitting-tensile strength of FW and SW concretes measured at (**a**) 28 days and (**b**) 91 days.

The replacement of FW with SW increased the 28 and 91 days of splitting-tensile strength of concrete by 24% and 13%, respectively. Adding SW to GGBFS-OPC-containing mixes could alter their hydration mechanism and result in improved splitting-tensile strength of concrete during both the early and late stages of concrete development. The enhanced splitting-tensile strength may be due to the reactions between $Cl^-$ and $SO_4^{-2}$ ions that are present in SW and the Al-rich phases found in GGBFS [57]. The presence of an alkaline environment is conducive to the occurrence of pozzolanic reactions, which are

beneficial for the GGBFS-OPC-containing mixes. Additionally, the incorporation of GGBFS could lead to a decrease in the formation of CH (portlandite), which is prone to leaching in the later stages of SW concrete development. Minimizing the amount of $Ca^{+2}$ ions present in the GGBFS-OPC-containing mixes can also reduce the formation and crystallization of expansive salts, such as gypsum, at later stages of SW concrete development.

Incorporating PP fiber into SW concrete is advantageous in managing the splitting-tensile strength at later stages of development. The addition of fibers creates a three-dimensional network that acts as a bridge to control the formation and spread of micro-cracks due to possible shrinkage and salt crystallization within the concrete matrix. Introducing PP fiber to plain concrete typically decreases leaching by decreasing the number of micro-cracks and the porosity of the material [63]. Therefore, the combination of fiber can effectively enhance the crack resistance and tensile strength of SW-mixed concrete. SW concrete mix containing 0.3% PP fiber yielded around 50% more splitting-tensile strength than plain FW concrete at both 28 and 91 days.

### 3.4. Flexural Strength

Figure 6 shows the FS of FW and SW concretes with 0%, 0.15%, and 0.3% PP fiber content. The incorporation of 0.15% and 0.3% PP fiber resulted in a 19.5% and 26% increase in the 28-day FS. While at 91 days, the FS of FW concrete was increased by 23.1% and 29% at the inclusion of 0.15% and 0.3% PP fiber, respectively. The incorporation of PP fiber into the mixture of plain concrete can effectively enhance its FS by strengthening its ability to resist cracking and increasing its ductility. By reinforcing the concrete with fibers, the tensile strength of the material is enhanced, decreasing the likelihood of cracking and improving its ability to withstand bending loads. Previous studies [61,64] have verified that the introduction of micro-fiber at the appropriate dosage into FW concrete can result in a 15–25% enhancement in its FS. The efficiency of PP fiber incorporation could improve with aging due to the strengthening and densification of the matrix.

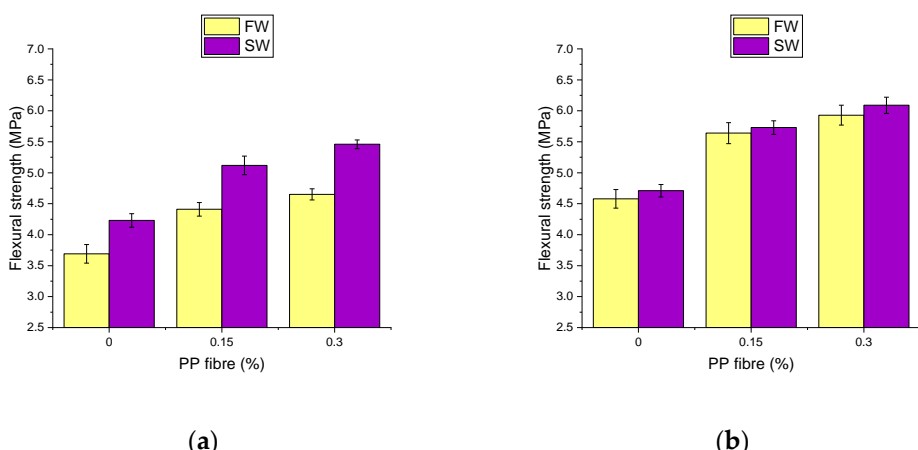

**Figure 6.** Effect of PP fiber on FS of FW and SW concretes measured at (**a**) 28 days and (**b**) 91 days.

Replacement of FW with SW resulted in improvements of 14.6% and 2.8% in the 28- and 91-day FS of concrete, respectively. The 28-day FS of the concrete made with hybrid OPC-GGBFS binder was enhanced by the accelerating effect of SW. At 91 days, the disparity in FS values between FW and SW was not significant (up to 3%). The study's findings demonstrated that similar to the other mechanical properties, FS of concrete could not be substantially affected by the addition of SW at later stages. PP fiber incorporation is beneficial in different aspects. It significantly advances the FS of SW and FW mixes, and it could also provide effective control on the loss in early age strength with the usage of OPC-GGBFS binder for SW applications. The incorporation of PP fiber could also advance the resistance against the micro-cracking caused by the crystallization of expansive salts in the matrix of the concrete. In terms of economical and life-cycle environmental impact, it is

more advantageous to use GGBFS-OPC binder mixes for SW mixes and marine applications. The incorporation of PP fiber could further enhance the mechanical strength, sustainability, and crack resistance of SW-mixed concrete.

### 3.5. Water Absorption Capacity

The effect of 0.15% and 0.3% PP fiber on the WA capacities of FW and SW concretes is illustrated in Figure 7. Concrete's durability and capability to withstand environmental factors, including moisture, freezing, and thawing cycles, and chemical erosion, are significantly affected by its WA capacity. The greater the concrete's WA, the more susceptible it becomes to damage from these external elements, leading to a decline in its strength and lifespan. The result shows that the addition of SW as a replacement for FW resulted in the decline of the WA capacity of concrete. Around 7–12% net reduction in the WA capacity of concrete was observed when the FW was replaced with SW. This reduction could be attributed to the accelerating effect of SW addition on the hydration and pozzolanic reactions in the GGBFS-OPC binder. SW could increase the hydration products leading to the densification of the concrete matrix, leading to reduced connectivity between pores [53,60].

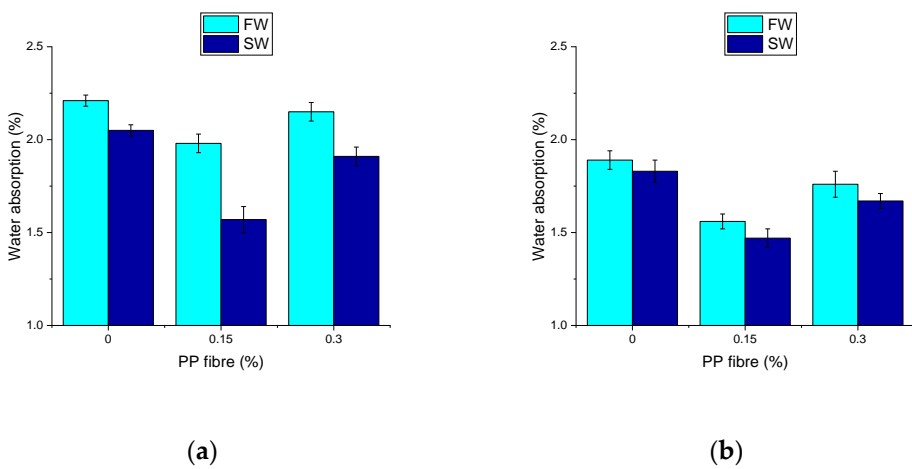

**Figure 7.** Effect of PP fiber on the WA capacity of FW and SW concretes measured at (**a**) 28 days and (**b**) 91 days.

The addition of PP fiber showed a positive impact on the WA resistance of both FW and SW concretes. The lowest WA values were noted for mixes incorporating 0.15% fiber content. Whereas for mixes containing 0.3% fiber, the WA value was slightly increased as compared to mixes incorporating 0.15% fiber. The addition of 0.15% PP fiber reduced the WA of FW concrete by 11% and 17% at 28 and 91 days, respectively. Whereas the addition of 0.3% PP fiber caused only a 3–7% net reduction in the WA capacity of FW concrete. Micro-fiber addition in concrete is believed to enhance its impermeability by controlling the development of micro-cracks that originate from the shrinkage or volumetric changes in the concrete's microstructure [65].

The SW mix experienced net reductions of around 20% and 9% with the addition of 0.15% and 0.3% of PP fiber, respectively. Therefore, PP fiber addition could improve the WA resistance of SW concretes. Fiber reinforcement in SW concrete could also assist in managing or delaying the degradation of its long-term impermeability caused by the formation of micro-cracks resulting from expansion due to salt crystallization. It is worth noting in Figure 7 that the difference between WA values of FW and SW concretes was reduced when the concrete aged between 28 and 91 days. This showed that the addition of SW could promote the growth and densification of microstructure at an early age, whereas FW concrete's microstructure continues to develop at a slower rate and takes a long time to reach maturity.

### 3.6. Electric Flux or Rapid Chloride Ion Permeability (CIP)

The CIP or charge passed through each concrete mix under standard conditions is illustrated in Figure 8. For mixes containing FW, CIP results can be used as an indicator of their corrosion resistance in marine environments. Normally, FW mixes with a low w/b ratio (<0.4) can be considered for steel-reinforced marine concrete structures [14]. Whereas SW concrete mixes which already contain $Cl^-$ ions are not suitable for steel-reinforced concrete, but can be considered for plain concrete construction or polymer fiber-rebar reinforced concretes [17]. For SW concretes, CIP test results can be related to the indication of permeability resistance or durability of concrete. The degree of permeability of $Cl^-$ is linked with the charge passed through the specimen, as shown in Table 5. ASTM C1202 [49] states that an CIP value of fewer than 1000 coulombs indicates a concrete with a "very low" permeability category, meaning that it is internally impervious or sealed, examples include reactive powder and ultra-high performance concretes. While the concrete mixes with CIP values between 1000 and 2000 coulombs correspond to a "low" permeability category, which is typically associated with concretes having low w/b ratios or traditional high-strength concrete. The CIP values of all FW and SW concrete mixes remained below 1000 coulombs, falling in the range of 300–700 coulombs. It is well-known that the CIP values of concrete containing hybrid OPC-GGBFS binder are significantly lower as compared to concrete mixes containing pure OPC-based binder. When GGBFS is added to concrete, there is a significant reduction in the concrete's CIP capacity. This reduction can be explained by several factors. First, the C-S-H gel in GGBFS-based binder has a high physical absorption capacity, which could enhance the concrete's ability to bind with Cl- ions [12]. Second, GGBFS contains a high proportion of Al-phases, which could contribute to the concrete's Cl-binding capacity [66]. Finally, incorporating smaller GGBFS particles could result in a more tortuous (twisted and turning) path for Cl- to travel through the concrete, slowing down their ingress into the concrete. A recent study has shown that the incorporation of GGBFS as a 45% replacement of OPC resulted in the improvement of electrical resistivity and corrosion risk of FW-mixed mortar [55].

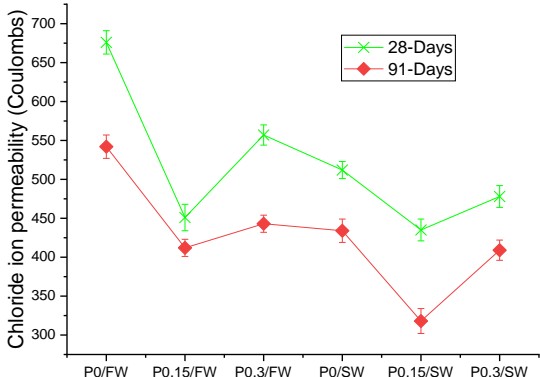

**Figure 8.** CIP values of FW and SW mixes with 0%, 0.15%, and 0.3% PP fiber.

**Table 5.** Relationship between level of $Cl^-$ permeability and charge passed [ASTM C1202 [49]].

| Permeability of $Cl^-$ | Charge (Coulombs) |
| --- | --- |
| High | Greater than 4000 |
| Moderate | 2000 to 4000 |
| Low | 1000 to 2000 |
| Very low | 100–1000 |
| Negligible | Smaller than 100 |

The FW replacement with SW resulted in around a 20% decline in the CIP capacity of concrete. Adding SW to hybrid OPC-GGBFS binder concrete may lead to an improvement

in resistance against chloride ingress. SW concrete mixes containing GGBFS have lower levels of CH contents, which could help in minimizing corrosion damage caused by excessive salt exposure owing to the pozzolanic reactions and pore refinement. Additionally, GGBFS has demonstrated an "interactive" or "synergistic" effect with SW that improves the chloride permeability resistance of concrete. This effect is due to the presence of SW salts, which accelerate the microstructural development of the concrete containing GGBFS.

PP fiber reinforcement had a notable improving effect on the CIP resistance of FW and SW concretes. In FW concrete, the incorporation of 0.15% and 0.3% PP fiber reduced the 28-day CIP value by 33% and 18%, respectively. Whereas the 28-day CIP value of SW concrete was reduced by 12% and 5% at 0.15% and 0.3% addition of PP fiber, respectively. It is known that fiber is effective in inhibiting the micro-cracking in concrete, it could be useful to control the ingress of fluids through the micro-cracks induced by shrinkage and crystallization in concrete matrix. Vafai et al. [57] reported the positive impact of polyvinyl fiber in reducing the porosity of concrete. The decrease in the total porosity could also contribute to the increased CIP resistance of SW and FW concretes. For both FW and SW groups, the lowest CIP values were observed at 0.15% content of PP fiber.

*3.7. Erosion Behavior under Simulated Tidal Environment (Drying–Wetting Behavior)*

The loss in mass in FW and SW concretes due to the simulated drying–wetting cycles is illustrated in Figure 9. The effect of PP fiber incorporation on the mass loss rate of FW and SW concretes after drying–wetting cycles is also shown in Figure 9. There was an initial negative mass loss (or mass loss) when drying–wetting cycles increased up to 48. For P0/FW, a maximum mass gain of 0.27% was noted at 28 cycles. Further increase in the drying–wetting cycle increased the mass loss, and up to 0.29% mass of FW concrete was lost after 126 cycles of drying–wetting. Whereas for P0/SW, the maximum mass gain of 0.29% was observed at 28 cycles, while after 126 cycles, the mass loss was found to be 0.56% (observed almost two times with FW concrete). In both P0/SW and P0/FW, the mass is gained due to ongoing hydration and pozzolanic reactions between residual unreacted cementitious materials. Therefore, both FW and SW concretes experienced net increases in the mass upon initial drying–wetting cycles. GGBFS helps in minimizing the drying–wetting erosion of concrete. First, the strength and density of concrete could increase in two ways when GGBFS is added to the mix. The ongoing pozzolanic reactions that occur with GGBFS can contribute to the increase in strength and intactness against drying–wetting erosion. Second, the filler effect of GGBFS particles can refine the pores in the concrete. Additionally, high temperatures during the drying could agitate secondary hydration in GGBFS-containing mixes, which could lead to the production of a denser microstructure and more cementitious gels, further enhancing the strength and density of the concrete [51].

Above 28 cycles, the FW and SW concrete mixes started to lose mass, owing to a series of internal and surface cracking under drying–wetting cycles. The increasing number of cracks in FW and SW concretes could promote mass loss. SW concrete mixes have lower erosion resistance as compared to FW mixes. This is due to the fact that SW concrete is prone to the higher production of expansive crystals of $CaCl_2$, which increases the hydration heat and micro-cracking. Ultimately, this could facilitate the transportation of harmful ions from the simulated water into the concrete matrix at later ages [67]. The creation of salts that could expand in SW concrete, such as gypsum, can potentially damage the strength of SW concrete mixes over time. This would make it easier for environmental SW to penetrate the SW concrete microstructure and result in physical and chemical damage.

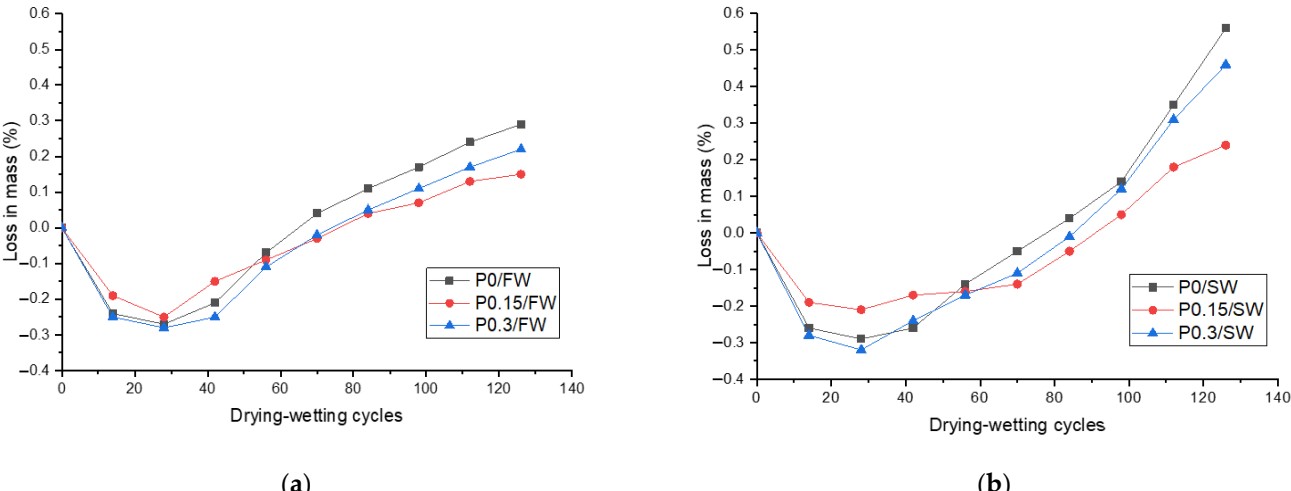

**Figure 9.** The drying–wetting induced mass loss of (**a**) FW and (**b**) SW concrete mixes with 0%, 0.15%, and 0.3% PP fiber contents.

After 56 cycles, when the plain SW and FW concrete matrices begin to erode, PP fiber reinforcement can have a positive impact on both internal and surface cracks in concrete. The incorporation of PP fiber also has a dominant effect on controlling the micro-cracks developed in the early stages [68]. Therefore, by providing more resistance to the ingress of harmful fluids, PP fiber can control the degradation of both FW and SW concretes at later ages. It was observed that 0.15% PP fiber incorporation could have a more beneficial effect on the erosion resistance of concretes as compared to 0.3% PP fiber. For instance, at 0.15% PP fiber, drying–wetting erosion resistance of FW and SW concretes was increased by 46% and 49%, respectively. Whereas at the incorporation of 0.3% PP fiber, the erosion resistance of FW and SW concretes was increased by 25% and 18%, respectively. The increase in fiber content could result in the clumping of fiber filaments, which may compromise the efficiency of fiber reinforcement in reducing the porosity of concrete, as compared to that observed with a low volume of fiber. The dispersion of fibers in concrete can be reduced if a large number of fibers are blended in, causing them to clump and entangle. If excessive fibers are incorporated, it can create additional channels for ions to penetrate the concrete, leading to a higher rate of fluid absorption–desorption during simulated drying–wetting cycles. Therefore, 0.15% PP fiber content could provide a more useful result in controlling tidal erosion of SW-mixed or FW-mixed concretes. Wang et al. [69] have also reported a significant control in the dry–wet cycles due to the addition of PP fiber in FW concrete.

## 4. Conclusions

In this study, the effect of PP fiber was studied on the mechanical and durability behavior of high-strength concrete mixes developed with freshwater (FW) and seawater (SW). The following are the key results:

1. PP fiber inclusion at 0.15% and 0.3% increased the compressive strength of SW concrete by up to 6–8%. Hybrid OPC-GGBFS binder mix could experience a 42% net increase in the 7 days of compressive strength due to the replacement of FW with SW.
2. The addition of PP fiber showed a promising effect on the splitting-tensile strength of SW concrete. SW concrete mix containing 0.3% PP fiber could yield around 50% more splitting-tensile strength than plain FW concrete at both 28 and 91 days.
3. No notable difference was observed between the effects of PP fiber addition on the flexural strength of FW and SW concretes. The incorporation of 0.3% PP fiber in SW concrete showed a net flexural strength improvement of around 30%, as compared to plain SW concrete.
4. The use of SW rather than FW also had a slightly positive impact on the 28- and 91-day WA capacity of concrete. At 91 days, SW concrete owing to the addition of

0.15% PP fiber attained 19% lower WA as compared to plain SW concrete and 22% lower WA as compared to plain FW concrete. The net reduction in WA observed with 0.15% PP fiber could be more than that noticed with 0.3% PP fiber.

5. The incorporation of PP fiber was effective in reducing the CIP of SW concretes. The use of 0.15% and 0.3% PP fiber reduced the CIP of SW concrete by 19% and 5%, respectively. The use of 0.15% is more effective than 0.3% PP fiber in controlling the CIP value of SW concrete. FW concrete prepared with or without PP fiber is deemed suitable for marine construction with steel reinforcement. The incorporation of PP fiber can further improve the corrosion resistance of FW concrete. The permeability of chlorides inside SW concrete with or without PP fiber was also "very low"; however, due to the presence of chlorides in SW concrete, it is suitable for plain concrete or fiber-reinforced polymer rebar concrete applications.

6. After 126 cycles, SW concrete yielded 0.56% mass loss as compared to 0.29% mass loss of FW concrete. The incorporation of PP fiber reinforcement is highly beneficial against the erosion of SW and FW concretes in a simulated tidal environment. The use of 0.15% and 0.3% PP fiber can reduce the erosion of SW concrete by almost 50% and 18%, respectively.

**Author Contributions:** Conceptualization, T.A. and B.A.; methodology, B.A. and H.A.; software, B.A.; validation, T.A., S.S.R. and M.H.; formal analysis, S.S.R. and H.A.; investigation, T.A.; resources, T.A.; data curation, T.A. and B.A.; writing—original draft preparation, T.A.; writing—review and editing, S.S.R.; visualization, B.A.; supervision, S.S.R. and H.A.; project administration, B.A. and M.H.; funding acquisition, T.A. All authors have read and agreed to the published version of the manuscript.

**Funding:** The authors would like to thank the Deanship of Scientific Research at Umm Al-Qura University for supporting this work by Grant Code: (23UQU4290255DSR008).

**Institutional Review Board Statement:** Not applicable.

**Informed Consent Statement:** Not applicable.

**Data Availability Statement:** Data will be made available upon request.

**Acknowledgments:** The authors would like to thank the Deanship of Scientific Research at Umm Al-Qura University for supporting this work by Grant Code: (23UQU4290255DSR008).

**Conflicts of Interest:** The authors declare no conflict of interest.

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
