# Peer review of "Investigating the Effects of Polypropylene Fibers on the Mechanical Strength, Permeability, and Erosion Resistance of Freshwater and Seawater Mixed Concretes"

_jmse, doi:10.3390/jmse11061224_

Round 1
Reviewer 1 Report
The manuscript entitled “Controlling the tidal erosion of seawater mixed concrete using low to medium volume of polymer fibre” is a paper of medium-low quality. It is interesting and deals with an important and recent problem of sustainable building materials, in this case seawater usage for concrete production and the mechanical and durability properties of concrete. The article requires amendments. I recommend it to be published after a major revision.
Main issues – the article requires improvement:
1. Line 194. For me it should be cured in SW also. You will obtain one more factor for analysis. Also if you discuss a marine structures it will be closer to reality! You should complete this research – it will significantly improve the manuscript. Even if it is time consuming, I think it is worth it.
2. Par 2.3.2. the problem with chlorides in your case is already inside concrete. The penetration of chloride ions from outside is not as important as prevention of steel corrosion by other means than not allowing chlorides enter the concrete. You have to advocate the choice of such method, and prove its efficiency.
3. Only section 3.7 is interesting. All other presented properties are predictable. It may be improved by presentation of SW cured samples.
4. Conclusions 1-3 are too obvious. 4,5 and 7 are ok. 6 sholud be advocated together with the method.
Minor comments:
5. Paragraph 1 – it should be indicated that it is similar effect to this caused by introducing accelerating admixture containing calcium chloride.
6. Lines 55-56: is it indeed an adverse effect? For me it depends on the situation.
7. Line 107 the word „concrete” is missing probably.
8. I know that the article is on concrete itself, but the chloride induced corrosion of reinforcing steel should be briefly presented as well.
9. Line 125 BFS is not a steel slug, Those are two different materials. BFS is from blast furnaces and the steel one in the converter process. Please be more accurate in this matter.
10. Lines 144, 145, 202 (maybe in other places as well) – space between number and unit is required.
11. Line 165 W/CM is not commonly used abbreviation. w/c should be used. w/b is ok.
12. Line 173 “a slump range of 150 to 250 mm was chosen” – this is quite big range. For further research I suggest to limit it to 150-200 or 200-250, or any other span not greater than 50 mm.
13. GGBS and BFS are used in the paper. One of it should be used.
14. I think that using abbreviations for flexural, tensile strength, compressive strength (FS, STS, CS) is not a good practice. You should use whole names.
Author Response
The manuscript entitled “Controlling the tidal erosion of seawater mixed concrete using low to medium volume of polymer fibre” is a paper of medium-low quality. It is interesting and deals with an important and recent problem of sustainable building materials, in this case seawater usage for concrete production and the mechanical and durability properties of concrete. The article requires amendments. I recommend it to be published after a major revision.
Response: We sincerely appreciate your time and effort in reviewing our paper. Your valuable insights and feedback have greatly enhanced the quality of our work. Thank you for your valuable contribution to our research.
Main issues – the article requires improvement.
Comment #1: Line 194. For me it should be cured in SW also. You will obtain one more factor for analysis. Also, if you discuss a marine structures, it will be closer to reality! You should complete this research – it will significantly improve the manuscript. Even if it is time consuming, I think it is worth it.
Response: Thank you for your valuable feedback. We acknowledge that conducting research on the properties of concrete cured in saltwater (SW) would have been beneficial. However, due to limitations in the number of samples and the completion of the project, it would take more than six months to complete such research. Nonetheless, we are currently engaged in a separate project that focuses on evaluating the performance of different fiber reinforced concretes with SW mixing and curing. Although the scope and testing methods differ from the present study, these findings will provide valuable insights. It is important to note that while SW curing aligns closely with marine structures, the results of our research on the effect of PPF on the performance of both SW and freshwater (FW) mixed concrete will have practical implications. Additionally, the findings related to the tidal erosion behavior can prove to be highly useful for marine structures.
Comment #2: Par 2.3.2. the problem with chlorides in your case is already inside concrete. The penetration of chloride ions from outside is not as important as prevention of steel corrosion by other means than not allowing chlorides enter the concrete. You have to advocate the choice of such method, and prove its efficiency.
Response (Lines 433-440): Thank you for addressing the concern regarding the relevance of CIP test results in the context of chloride presence in seawater (SW) concrete. The CIP test outcomes can indeed serve as a valuable tool for evaluating the corrosion resistance of freshwater (FW) concrete mixes in marine environment. However, when it comes to SW concrete, the CIP test can be regarded as an indicator of the concrete's resistance to permeability. The permeability can be related with the erosion resistance and durability of SW and FW concretes. In the case of SW mixed concretes, it is inevitable to prevent steel corrosion. Hence, SW mixing is primarily advantageous for applications involving plain concrete and for structural concrete reinforced with polymer bars.
Comment #3: Only section 3.7 is interesting. All other presented properties are predictable. It may be improved by presentation of SW cured samples.
Response: Please see the response to comment 1.
Comment #4: Conclusions 1-3 are too obvious. 4,5 and 7 are ok. 6 sholud be advocated together with the method.
Response (Lines 564-572): We have revised the conclusion considering that the CIP assessment. Its usefulness has been linked with the permeability-resistance of concrete. Earlier, we discussed the effect of PF content on the RCIP resistance. The results are quite useful, since very few studies have investigated the effects of polymer fibers on the CIP properties of SW concrete.
Minor comments
Comment #5: Paragraph 1 – it should be indicated that it is similar effect to this caused by introducing accelerating admixture containing calcium chloride.
Response (Lines 58-59): This statement has been added.
Comment #6: Lines 55-56: is it indeed an adverse effect? For me it depends on the situation.
Response (Lines 63): We have revised the wording. The wording is changed to declining effect instead of adverse effect. Thank you for these invaluable insights.
Comment #7: Line 107 the word „concrete” is missing probably.
Response (Line 124): Added. Thank you for these corrections.
Comment #8: I know that the article is on concrete itself, but the chloride induced corrosion of reinforcing steel should be briefly presented as well.
Response (Line 67-75): We have added a new paragraph relating to the discussion corrosion induced by chlorides.
Comment #9: Line 125 BFS is not a steel slug, those are two different materials. BFS is from blast furnaces and the steel one in the converter process. Please be more accurate in this matter.
Response (Line 142-143): Thank you for your insights. We have replaced the term with BFS with GGBFS throughout the paper and revised these lines. The slag was obtained from blast furnace slag.
Comment #10: Lines 144, 145, 202 (maybe in other places as well) – space between number and unit is required.
Response (Line 164. 165): We have inserted the spaces between the quantities and units.
Comment #11: Line 165 W/CM is not commonly used abbreviation. w/c should be used. w/b is ok.
Response (Line 185): Thank you for this correction. We have replaced W/CM with w/b. throughout the paper.
Comment #12: Line 173 “a slump range of 150 to 250 mm was chosen” – this is quite big range. For further research I suggest to limit it to 150-200 or 200-250, or any other span not greater than 50 mm.
Response: Dear reviewer, thank you for sharing this invaluable information and suggestion. We will consider suggested ranges in future research.
Comment #13: GGBS and BFS are used in the paper. One of it should be used.
Response: Complied. GGBFS is used throughout the paper.
Comment #14: I think that using abbreviations for flexural, tensile strength, compressive strength (FS, STS, CS) is not a good practice. You should use whole names.
Response: Dear reviewer, we appreciate your feedback. We acknowledge that including full names can be beneficial for readers. However, using the complete forms of the names often results in overly long or wordy sentences. Hence, we have chosen to utilize abbreviations instead. Additionally, repeatedly mentioning the full name of the property in subsequent sentences can become monotonous, hence our preference for abbreviations.
Reviewer 2 Report
An experimental study was conducted to determine the effect of polypropylene (PP) fiber on seawater-mixed concrete's fresh and hardened properties. Although the results are interesting, a major revision is necessary to improve the manuscript, as follows:
- Abstract: Please rewrite the abstract, Lines 27-29: effect of GGBFS on SW concrete is an objective of this study? Why is this point not mentioned in the title?
- What is the difference between GGBFS and BFS?
- What is polymer fibre? The authors need to explain more about experimental details in the abstract.
- Abstract: Please add some quantitative results at the end of the abstract.
- Page 1, Line 41: please use “freshwater (FW)” instead of FW in this sentence.
- The reviewer recommends changing the title to: Synergistic effects of polymer fiber and ground blast furnace slag to mitigate the tidal erosion of seawater mixed concrete”
- The reviewer recommends using “polypropylene (PP) fiber” instead of polymer fiber or PF throughout the manuscript.
- Figure 2: please use scale bar for this figure.
- Page 6, Line 197: please cross-check the table numbering throughout the manuscript. This should be Table 4.
- Page 6, Line 197: What is “Effective w/b” in this table?
- General point: Results of Figure 3 is interesting and valuable for researchers. However, the authors should consider this point that mechanical properties of mixtures with the same range workability can be compared; for instance: mixtures FM mixed and SM mixed (0.45% SP) can be compared for the mechanical properties only, while using SM mixed (0.3% SP) is not accurate.
- Page 7, Line 266: GGBS or BFS? The authors should use constant abbreviations throughout the manuscript.
- Page 15, Line 506: what is “polypropylene/polymer fibre (PF)”?
- Regarding GGBS, the reviewer recommends comparing your results with: “Bhojaraju, C., Mousavi, S. S., & Ouellet-Plamondon, C. M. (2023). Influence of GGBFS on corrosion resistance of cementitious composites containing graphene and graphene oxide. Cement and Concrete Composites, 135, 104836.”
Minor editing of English language required.
Author Response
An experimental study was conducted to determine the effect of polypropylene (PP) fiber on seawater-mixed concrete's fresh and hardened properties. Although the results are interesting, a major revision is necessary to improve the manuscript, as follows:
Response: We would like to express our sincere gratitude for reviewing our paper. Your input has been instrumental in refining our ideas and strengthening the overall impact of our study.
Comment #1: Abstract: Please rewrite the abstract, Lines 27-29: effect of GGBFS on SW concrete is an objective of this study? Why is this point not mentioned in the title?
Response (L27-29): Dear reviewer, the article investigates the performance of FW and SW concretes with different contents of PP fiber. We did not specifically study the effect of GGBFS incorporation on the properties of concrete. We have further specified the title of our research work. The new title is “Effect of polypropylene fibre on the mechanical, permeability and erosion-related properties of seawater mixed concrete”.
Comment #2: What is the difference between GGBFS and BFS?
Response: We have replaced the term BFS with GGBFS to be more accurate.
Comment #3: What is polymer fibre? The authors need to explain more about experimental details in the abstract.
Response (Line 43): We have replaced the name of the fibre with PP fibre (Polypropylene fibre) throughout the paper and figures.
Comment #4: Abstract: Please add some quantitative results at the end of the abstract.
Response (Line 32-36; and 38-40): Quantitative information has been incorporated into the abstract section.
Comment #5: Page 1, Line 41: please use “freshwater (FW)” instead of FW in this sentence.
Response: Corrected.
Comment #6: The reviewer recommends changing the title to: Synergistic effects of polymer fiber and ground blast furnace slag to mitigate the tidal erosion of seawater mixed concrete”.
Response: Please see the response to comment 1.
Comment #7: The reviewer recommends using “polypropylene (PP) fiber” instead of polymer fiber or PF throughout the manuscript.
Response: Complied. Thanks for this suggestion.
Comment #8: Figure 2: please use scale bar for this figure.
Response: The scale has been added in this picture.
Comment #9: Page 6, Line 197: please cross-check the table numbering throughout the manuscript. This should be Table 4.
Response: Thank you for this important correction. The table numbers have been corrected.
Comment #10: Page 6, Line 197: What is “Effective w/b” in this table?
Response: Thank you for this correction. It was meant to be water column.
Comment #11: General point: Results of Figure 3 is interesting and valuable for researchers. However, the authors should consider this point that mechanical properties of mixtures with the same range workability can be compared; for instance: mixtures FM mixed and SM mixed (0.45% SP) can be compared for the mechanical properties only, while using SM mixed (0.3% SP) is not accurate.
Response (See Table 4): Dear reviewer the properties of FW and SW mixed concretes were compared for the same workability. Therefore, SW concretes were prepared with 0.45% SP and FW concretes were prepared with 0.3% SP. Please see Table 4, where SW concrete mixes have increased SP dosage to attain the same workability as FW concretes. All hardened state properties were evaluated for mixed given in Table 4.
Comment #12: Page 7, Line 266: GGBS or BFS? The authors should use constant abbreviations throughout the manuscript.
Response: The term GGBFS has been used throughout the paper. Thank you, for these important corrections.
Comment #13: Page 15, Line 506: what is “polypropylene/polymer fibre (PF)”?
Response: We have replaced the term ‘polymer fiber’ with polypropylene fiber (PP Fibre) to be more specific.
Comment #14: Regarding GGBS, the reviewer recommends comparing your results with: “Bhojaraju, C., Mousavi, S. S., & Ouellet-Plamondon, C. M. (2023). Influence of GGBFS on corrosion resistance of cementitious composites containing graphene and graphene oxide. Cement and Concrete Composites, 135, 104836.”
Response (Line 282-285; 454-456): Dear reviewer, thank you for recommending this new study on the GGBFS incorporation in cementitious materials. We have included this study into the discussion of our results.
Round 2
Reviewer 1 Report
The authors did not provide convincing arguments and did not improve the article enough for me to change my mind.
I understand that the results can also be applied in a freshwater (FW) environment, but it is primarily about seawater. Even the title of the journal should indicate this.
Still only subsection 3.7 is interesting. The others are predictable.
I accept the argumentation regarding CIP.
The conclusions need reconstruction. Certainly the first 3 are unnecessary and obvious. The use of abbreviations FS, STS and CS is unacceptable. It makes the article hard to read.
Author Response
Replies to reviewers’ comments.
Reviewer #1
The authors did not provide convincing arguments and did not improve the article enough for me to change my mind.
Response: Dear reviewer, thank you again for taking the time to provide feedback on our research paper. We appreciate your engagement with our work, and we value the opportunity to address your concerns. We understand that you found our arguments to be unconvincing and that you feel the article has not undergone significant improvements. We apologize if our work did not meet your expectations. We mentioned that an additional experimental program regrading the testing of mixes subjected to seawater curing is quite impossible at this stage of the project. We will try to cover more novel aspects of SW mixed and SW cured mixes in the future projects.
Comment 1: I understand that the results can also be applied in a freshwater (FW) environment, but it is primarily about seawater. Even the title of the journal should indicate this.
Response: We revised the title of our research work. The new title is “Investigating the effects of polypropylene fibres on the mechanical strength, permeability, and erosion resistance of freshwater and seawater mixed concretes”.
Comment 2: Still only subsection 3.7 is interesting. The others are predictable.
Response: Dear reviewer, the overall literature is deficient in studies related to the performance of fiber reinforced mixes prepared with seawater. Therefore, we believe that the other sections (other than tidal erosion) would also be beneficial for researchers and literature. The following high-quality research has also focussed on the strength and ductility parameters of steel fiber reinforced seawater mixed concrete.
https://www.sciencedirect.com/science/article/pii/S2352710223010021#sec3.
Comment 3: I accept the argumentation regarding CIP.
Response: Thank you, we learnt a lot from your previous feedback and extensive experience in the corrosion-related research.
Comment 4: The conclusions need reconstruction. Certainly, the first 3 are unnecessary and obvious. The use of abbreviations FS, STS and CS is unacceptable. It makes the article hard to read.
Response: We have removed the 1st conclusion (relating to workability) and shortened the 2nd and 3rd conclusion points (relating to CS and STS). We have also removed the abbreviations for FS, STS, and CS throughout the paper.
Reviewer 2 Report
The authors appropriately improved the manuscript structure.
Minor editing of English language required.
Author Response
Thank you for accepting the changes/revisions in the first round.